# Topology-Guided Graph Pre-training and Prompt Learning on Directed Graphs

**Peiyu Liang**                                             *peiyu.liang@temple.edu*
*Temple University*

**Chenguang Yang**                                          *cyang314@ucr.edu*
*University of California, Riverside*

**Yixuan He**                                               *Yixuan.He@asu.edu*
*Arizona State University*

**Rong Pan**                                                *Rong.Pan@asu.edu*
*Arizona State University*

**Yuzhou Chen**                                             *yuzhou.chen@ucr.edu*
*University of California, Riverside*

**Reviewed on OpenReview:** *https://openreview.net/forum?id=kMIdkLTys8*

## Abstract

In recent years, graph neural networks (GNNs) have been the dominant approach for graph representation learning, leading to new state-of-the-art results on many classification and prediction tasks. However, they are limited by the fact that they cannot effectively learn expressive node representations without the guide of labels, thus suffering from the labeled data scarcity problem. To address the challenges of labeling costs and improve robustness in few-shot scenarios, pre-training on self-supervised tasks has garnered significant attention. Additionally, numerous prompting methods have been proposed as effective ways to bridge the gap between pretext tasks and downstream applications. Although graph pre-training and prompt tuning methods have explored various downstream tasks on undirected graphs, directed graphs have been largely under-explored, and these models suffer limitations in capturing directional and topological information in directed graphs. In this paper, we propose a novel topology-guided directed graph pre-training and prompt tuning model, named TopoDIG, that can effectively capture intrinsic directional structural and local topological features in directed graphs. These features play essential roles in transferring knowledge from a pre-trained model to downstream tasks. TopoDIG consists of an encoder in the form of a magnetic Laplacian matrix, a topological encoder, and a graph prompt learning function. Experimental results on both real-world and synthetic directed graphs demonstrate the superior performance of TopoDIG compared to prominent baseline methods.

## 1 Introduction

With the growing prevalence of applications where data originate from non-Euclidean domains and are naturally represented as graphs—such as social networks, citation networks, and biochemical structures—graph data, rich in relational information, play a crucial role in numerous learning tasks (Zhou et al., 2020; He et al., 2024b). These tasks include predicting and modeling social interactions, protein interfaces, classifying diseases, learning molecular fingerprints, and modeling physical systems (Zhou et al., 2020; He et al., 2025). In many cases, these relationships inherently exhibit a sense of direction. For instance, the WebKB dataset

(Pei et al., 2020) consists of university websites connected by hyperlinks. In this setting, one website may link to another without necessarily receiving a reciprocal link. Such datasets are naturally represented by directed graphs/networks. Directed networks, with asymmetric sending and receiving patterns, are important with many applications, such as clustering time-series data with lead-lag relationships (He et al., 2022b), ranking from pairwise comparisons (He et al., 2022a), angular synchronization (He et al., 2024a), detecting influential groups in social networks (He et al., 2022b; Zhang et al., 2021b), and IIoT-based cognitive manufacturing (Liu et al., 2022a). Indeed, He et al. (2022b) points out that even in the absence of any edge density differences, directionality (i.e., edge orientation) can reveal latent properties of network flows. Moreover, Zhang et al. (2021b) introduces MagNet, a spectral Graph Neural Network (GNN) designed for directed graphs. MagNet is built on the magnetic Laplacian, a complex Hermitian matrix that captures both geometric and directional structure: the magnitude of its entries encodes undirected geometric relationships (here we refer to connection patterns), while their phase represents directional information (e.g., how information flows on a graph, or source and target information for edges). A tunable "charge" parameter is introduced, which asymptotically allows control over the emphasis on directionality for small parameter values. More generally, different charge parameter values induce sensitivity to different directed subgraph motifs. Although these models achieve promising results in the directed graph domain, they cannot simultaneously capture directional and topological structures which may result in the loss of important higher-order directional information (which is crucial for certain tasks).

Besides, recent trends in graph transfer learning have led to a proliferation of studies that learn useful graph representations that can be applied to various downstream tasks or different domains. Among transfer learning methods, graph prompt learning is a major area of interest and a number of recent efforts have been made on graph pre-training, graph prompt design, and graph fine-tuning (Sun et al., 2023b). For example, GPPT (Sun et al., 2022) employs edge prediction as a pre-training objective and redefines node classification within this framework by introducing labeled tokens into the original graph and aligning the classification task with the pretext objective. GraphPrompt (Liu et al., 2023) unifies pre-training and downstream tasks within a common task template while incorporating a learnable prompt that guides the downstream task in identifying the most relevant knowledge from the pre-trained model in a task-specific manner. All-in-One (Sun et al., 2023a) introduces a meta-learning technique to the design of graph prompt which improve multi-task performance. Nonetheless, most graph prompt learning do not take directed graph structure into account or may discard this salient information, e.g., Zi et al. (2024) lists six different pre-training methods and 5 graph prompt learning methods, however, none of these methods has been applied to directed graphs.

To address those two issues, in this work, we first propose a **Topology-Guided Directed Graph Pre-training and Prompt Learning (TopoDIG)** approach on directed graphs which learn from directional and topological information generated by magnetic Laplacian-based graph convolutional networks and Dowker complex-based topological representation learning module, respectively. In essence, our main contributions can be summarized as follows:

- We propose an innovative directed graph pre-training framework, which brings the concepts of magnetic Laplacian, Dowker complex-based topological features, and topological representation learning to the directed graph domain.

- We design a topology-empowered graph prompt function that improves the transfer and generalization capabilities of GNN.

- We present extensive experimental studies on both real-world and synthetic datasets and find that our proposed TopoDIG outperforms the competitive baselines in various few-shot node classification tasks. These observations demonstrate the effectiveness of our framework in practical applications.

## 2 Related Work

### 2.1 Graph Neural Networks for Directed Graphs

Graph Neural Networks (GNNs) are specifically designed to learn node representations by leveraging neural networks that operate directly on graph structures. Among the various deep learning approaches, the

message-passing framework has emerged as the dominant paradigm in recent works, including models such as Graph Convolutional Networks (GCN) (Kipf & Welling, 2017), GraphSAGE (Hamilton et al., 2017), and Graph Attention Networks (GAT) (Veličković et al., 2018). Through iterative aggregation steps, these methods enable each node's representation to incorporate information from its neighboring nodes across multiple hops which is crucial for enhancing the performance of downstream tasks. However, in directed graphs, these traditional GNNs struggle to fully capture the directional relationships between nodes. That is, the inherent symmetry of message-passing methods often disregards the edge direction, leading to a loss of important structural information. This limitation becomes particularly pronounced in tasks where the directionality of the graph plays a critical role such as in the analysis of causal relationships or in recommendation systems. To relieve this limitation, recently, several methods focus on handling directional structure during the propagation and leverage the power of directed Laplacian to uncover complex patterns. For instance, DGCN (Tong et al., 2020b) leverages first- and second-order proximity by constructing three Laplacians, but it can be inefficient in terms of space and computation speed. DiGCN (Tong et al., 2020a) simplifies DGCN by introducing a directed Laplacian based on PageRank and aggregating information while considering higher-order proximity. MagNet (Zhang et al., 2021b) builds a complex Hermitian matrix where the magnitude of its entries encodes undirected geometric structure, and their phase captures directional information. Tong et al. (2021) proposes a directed network data augmentation technique, Laplacian perturbation, and applies it to contrastive learning in directed graphs. DiGCL (He et al., 2022b) focuses on directed network clustering, introducing novel imbalance objectives and evaluation metrics based on flow imbalance measures.

## 2.2 Graph Pre-Training and Prompt Tuning

Most transfer learning approaches in graph representation learning follow the "pre-train & fine-tune" paradigm. This framework involves leveraging readily available information through a pretext task (e.g., node-level, edge-level, or graph-level) to learn meaningful representations which are subsequently fine-tuned on a downstream task using the pre-trained model as initialization. The primary objective of graph pre-training is to capture structural patterns from input graphs in a self-supervised manner. In general, pre-training methods can be categorized into three approaches based on their tailored learning architectures, i.e., node-level, edge-level, and graph-level respectively. Significant advancements have been made in each category. For the graph-level learning, BRep-BERT (Lou et al., 2023) integrates GNNs with Transformers and incorporates graph structural information into the Transformer to learn both global and local entity feature representations. At the edge-level, EdgePreGPPT (Sun et al., 2022) estimates link probabilities between node pairs using dot product calculations, and enhances the similarity of contextual subgraphs of linked pairs and reduces similarity for unlinked pairs by sampling triplets from label-free graphs. At the node-level, MoAMa (Inae et al., 2023) introduces a novel node-masking technique that enables the model to capture long-range inter-motif structures for graph pre-training. Additionally, the adaptation of knowledge from an unlabeled graph to a target downstream task is typically achieved through fine-tuning where the pre-trained GNN is refined using a limited amount of labeled data (Lu et al., 2021). Furthermore, graph prompt functions mandate a pre-training task that can be readily emulated and integrate the pre-trained model into downstream tasks smoothly. ProG (Zi et al., 2024) provides a comprehensive overview of 5 state-of-the-art graph prompt techniques from GPPT (Sun et al., 2022) to GPF-plus (Fang et al., 2023) to All-in-one (Sun et al., 2023a). The fine-tuning strategy is inherently related to the pre-training method. Specifically, if pre-training is conducted at the graph-level, an appropriate graph-level fine-tuning approach should be employed, e.g., S2PGNN (Lu et al., 2021) provides an adaptive fine-tuning framework for preserving the global information that optimally tailors the fine-tuning process to the characteristics of the pre-trained GNN and the downstream dataset in graph-level tasks. However, all these approaches fall short in their ability to incorporate directional structure and topological information, the essence of directed graphs.

## 2.3 Deep Learning with Topology

Topological data analysis (TDA) (Wasserman, 2018; Chazal & Michel, 2021; Carlsson et al., 2012; Chen & Gel, 2025; Dixon et al., 2025; Chen et al., 2021b), a collection of methods derived from algebraic topology, has demonstrated significant utility across various machine learning tasks due to its robustness to noise and adaptability to diverse data modalities, including images, time series, and graphs. Among TDA techniques,

persistent homology (PH) has gained prominence in image classification, graph learning, and text mining by capturing topological structures across multiple intensity levels. Additionally, approaches such as discrete Morse theory, topological interactions, and center-line transforms have further contributed to performance improvements in these domains. In graph learning tasks, TDA has been leveraged in graph neural networks for topology learning by incorporating topological descriptor embeddings or PH-based loss functions to enhance GNN's performance (Carrière et al., 2020; Chen et al., 2021a; Zhao & Wang, 2019; Arafat et al., 2025; Horn et al., 2022; Chen et al., 2025b;a; 2024). Despite these advancements, conventional persistent homology pipelines often face substantial scalability challenges when applied to large-scale graphs. Standard constructions such as Vietoris–Rips or clique complexes typically incur prohibitive computational and memory costs, as the number of simplices grows exponentially with graph density and size. To address this issue, new approaches based on weaker complexes such as Dowker complex which can effectively reduce computational complexity (Dowker, 1952; Liu et al., 2022b; Choi et al., 2024; Li et al., 2025), which constitute a promising research direction. However, prior research primarily concentrate on undirected graphs, and Dowker complex has untapped potential in transfer learning for directed graphs where learning meaningful representations across tasks remains challenging. Our work, TopoDIG, introduces a novel framework that integrates directed geometric structure learning, topology-level information learning, graph prompting, and fine-tuning to facilitate knowledge transfer in the directed graph domain. More specifically, by leveraging PH to extract topological features, our approach enhances the adaptability of pre-trained models while preserving critical directional structural properties. This represents a significant departure from previous applications of TDA and extends its impact beyond unsupervised topology discovery to informed transfer learning across complex graph domains.

## 3 Method

**Problem Definition.** Denote a (possibly weighted) directed graph (digraph) with node attributes as $\mathcal{G} = (\mathcal{V}, \mathcal{E}, w, \mathbf{X})$, with $\mathcal{V}$ the set of nodes, $\mathcal{E}$ the set of directed edges/links, and $w \in [0, \infty)^{|\mathcal{E}|}$ the set of edge weights. $\mathcal{G}$ may have self-loops, but no multiple edges. The number of nodes is $n = |\mathcal{V}|$, and $\mathbf{X} \in \mathbb{R}^{n \times d_{\text{feat}}}$ is a matrix whose rows encode node features. This network can be represented by the node feature matrix $\mathbf{X}$ and the adjacency matrix $\mathbf{A} = (A_{ij})_{i,j \in \mathcal{V}}$, with $\mathbf{A}_{ij} = 0$ if no edge exists from $v_i$ to $v_j$; if there is an edge $e$ from $v_i$ to $v_j$, we set $A_{ij} = w_e$, the edge weight.

In this section, we will introduce, in detail, the TopoDIG framework, which leverages directional structural features and topological information of the directed graph to enhance the expressiveness and generalization capability of GNNs. TopoDIG mainly comprises three modules: magnetic Laplacian-based GNN module, topological signature modeling module, and graph prompt learning module. Figure 1 provides an intuitive illustration of the design of pre-training and prompting framework (including magnetic Laplacian-based GNN module and topological signature modeling module) within TopoDIG.

### 3.1 Magnetic Laplacian-Based Graph Neural Networks

MagNet (Zhang et al., 2021b) applies the concept of a magnetic Laplacian to directed graph neural networks, building on a parameterized family of magnetic Laplacians (Fanuel et al., 2017; F. de Resende & F. Costa, 2020). Our spectral GNN part is built upon MagNet. We first define a symmetrized adjacency matrix and a corresponding degree matrix as follows:

$$\tilde{\mathbf{A}}_{i,j} := \frac{1}{2}(\mathbf{A}_{i,j} + \mathbf{A}_{j,i}), \;\; 1 \le i, j \le n,$$

$$\tilde{\mathbf{D}}_{i,i} := \frac{1}{2}\sum_{j=1}^{n}(\mathbf{A}_{i,j} + \mathbf{A}_{j,i}), \;\; 1 \le i \le n,$$

with $\tilde{\mathbf{D}}_{i,j} = 0$ for $i \ne j$. To incorporate directional information, Zhang et al. (2021b) introduces a phase matrix $\mathbf{\Theta}^{(q)}$ by $\mathbf{\Theta}_{i,j}^{(q)} := 2\pi q(\mathbf{A}_{i,j} - \mathbf{A}_{j,i})$, where $q \in \mathbb{R}$ is the so-called "charge parameter". Using elementwise multiplication (denoted by $\odot$) and the imaginary unit i, we define a complex Hermitian matrix as $\mathbf{H}^{(q)} := \tilde{\mathbf{A}} \odot \exp(\mathrm{i}\mathbf{\Theta}^{(q)})$, where $\exp(\mathrm{i}\mathbf{\Theta}^{(q)})$ is applied elementwise by $\exp(\mathrm{i}\mathbf{\Theta}^{(q)})_{i,j} := \exp(\mathrm{i}\mathbf{\Theta}_{i,j}^{(q)})$.

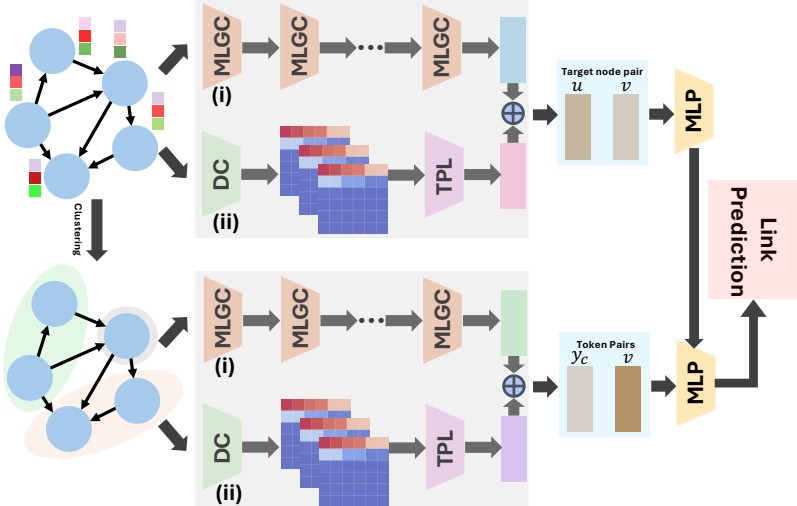

Figure 1: Overview of the TopoDIG pre-training and prompting framework. Given an input directed graph with node attributes, TopoDIG learns transferable representations through two complementary pathways (with the gray box). The first branch (i) applies magnetic Laplacian–based graph convolution (MLGC) layers to capture directional and local structural information. In parallel, the second branch (ii) constructs a Dowker complex to encode higher-order relational structures, which are transformed into compact topological embeddings via the topological representation learning (TPL) module. The structural and topological representations are fused to form joint node embeddings. Pre-training is performed using a pairwise prediction objective on both node-node and token-node pairs with a shared MLP and pretext loss, enabling topology-aware and prompt-compatible representations for downstream tasks.

Since $\tilde{\mathbf{A}}$ is symmetric and $\mathbf{\Theta}^{(q)}$ is skew-symmetric, $\mathbf{H}^{(q)}$ is Hermitian. Notably, setting $q = 0$ results in $\mathbf{H}^{(0)} = \tilde{\mathbf{A}}$, effectively symmetrizing the input graph and removing directional information.

Given $\mathbf{H}^{(q)}$, we define the unnormalized and normalized magnetic Laplacians as follows

$$\mathbf{L}_U^{(q)} \coloneqq \tilde{\mathbf{D}} - \mathbf{H}^{(q)} = \tilde{\mathbf{D}} - \tilde{\mathbf{A}} \odot \exp(\mathrm{i}\mathbf{\Theta}^{(q)}),$$

and

$$\mathbf{L}_N^{(q)} \coloneqq \mathbf{I} - \left( \tilde{\mathbf{D}}^{-1/2} \tilde{\mathbf{A}} \tilde{\mathbf{D}}^{-1/2} \right) \odot \exp(\mathrm{i}\mathbf{\Theta}^{(q)}) \,.$$

These Laplacian matrices are Hermitian, positive-definite, and the eigenvalues of $\mathbf{L}_N^{(q)}$ lie in $[0, 2]$.

Given $\mathbf{L}$ (one of the Laplacian matrices, in our experiments, the normalized one), let $\mathbf{u}_1 \ldots, \mathbf{u}_n$ be an orthonormal basis of eigenvectors satisfying $\mathbf{L}\mathbf{u}_k = \lambda_k \mathbf{u}_k$. Let $\mathbf{U}$ be a matrix whose $k$-th column is $\mathbf{u}_k$, for $1 \le k \le n$. The Fourier transform of a signal $\mathbf{x} : \mathcal{V} \to \mathbb{C}$ is defined as $\widehat{\mathbf{x}}(k) = \langle \mathbf{x}, \mathbf{u}_k \rangle \coloneqq \mathbf{u}_k^\dagger \mathbf{x}$, and equivalently, $\widehat{\mathbf{x}} = \mathbf{U}^\dagger \mathbf{x}$. Since $\mathbf{U}$ is unitary, we obtain the Fourier inversion formula

$$\mathbf{x} = \mathbf{U}\widehat{\mathbf{x}} = \sum_{k=1}^{n} \widehat{\mathbf{x}}(k)\mathbf{u}_k \,. \tag{1}$$

The convolution of $\mathbf{x}$ with a filter $\mathbf{y}$ is defined as the elementwise multiplication in the Fourier domain: $\widehat{\mathbf{y} * \mathbf{x}}(k) = \widehat{\mathbf{y}}(k)\widehat{\mathbf{x}}(k)$. By equation 1, this implies $\mathbf{y} * \mathbf{x} = \mathbf{U}\mathrm{Diag}(\widehat{\mathbf{y}})\widehat{\mathbf{x}} = (\mathbf{U}\mathrm{Diag}(\widehat{\mathbf{y}})\mathbf{U}^\dagger)\mathbf{x}$, where $\mathrm{Diag}(\mathbf{z})$ denotes a diagonal matrix with the vector $\mathbf{z}$ on its diagonal. Thus, we define a *generalized convolution matrix* $\mathbf{Y}$ as

$$\mathbf{Y} = \mathbf{U}\mathbf{\Sigma}\mathbf{U}^\dagger \,, \tag{2}$$

for a diagonal matrix $\mathbf{\Sigma}$. This generalizes the spectral convolutions introduced in Bruna et al. (2014).

Following Zhang et al. (2021b), we approximate spectral convolutions using polynomials of $\mathbf{L}$. We set $\boldsymbol{\Sigma} = \sum_{k=0}^{K} \theta_k T_k(\widetilde{\boldsymbol{\Lambda}})$ as a polynomial in $\mathbf{L}$, using Chebyshev polynomials $T_k(\cdot)$, where $\widetilde{\boldsymbol{\Lambda}} = \frac{2}{\lambda_{\max}}\boldsymbol{\Lambda} - \mathbf{I}$ normalizes eigenvalues to $[-1, 1]$. For $0 \leq k \leq K$, $T_k$ is the Chebyshev polynomials defined by $T_0(x) = 1, T_1(x) = x$, and $T_k(x) = 2xT_{k-1}(x) + T_{k-2}(x)$ for $k \geq 2$. Since $\mathbf{U}$ is unitary, we have $(\mathbf{U}\widetilde{\boldsymbol{\Lambda}}\mathbf{U}^\dagger)^k = \mathbf{U}\widetilde{\boldsymbol{\Lambda}}^k\mathbf{U}^\dagger$, and thus, letting $\widetilde{\mathbf{L}} := \frac{2}{\lambda_{\max}}\mathbf{L} - \mathbf{I}$, we have

$$\mathbf{Y}\mathbf{x} = \mathbf{U}\sum_{k=0}^{K} \theta_k T_k(\widetilde{\boldsymbol{\Lambda}})\mathbf{U}^\dagger\mathbf{x} = \sum_{k=0}^{K} \theta_k T_k(\widetilde{\mathbf{L}})\mathbf{x}. \tag{3}$$

Let $\mathbf{Z}_{\mathcal{G}}^{(l)}$ be the input matrix at layer $l$ (with $\mathbf{Z}_{\mathcal{G}}^{(0)} = \mathbf{X}$ being an $n \times F_0$ input matrix with columns $\mathbf{z}_{\mathcal{G},1}^{(0)}, \ldots \mathbf{z}_{\mathcal{G},F_0}^{(0)}$), and $L$ denotes the number of convolution layers. As in Zhang et al. (2021b), we use a complex activation function defined by $\sigma(q) = q$, if $-\pi/2 \leq \arg(q) < \pi/2$, and $\sigma(q) = 0$ otherwise, where $\arg(\cdot)$ is the complex argument of $q \in \mathbb{C}$. Let $F_\ell$ be the number of channels in the $\ell$-th layer. For $1 \leq \ell \leq L$, $1 \leq i \leq F_\ell$, and $1 \leq j \leq F_{\ell-1}$, let $\mathbf{Y}_{ij}^{(\ell)}$ be a convolution matrix defined by Eq. equation 2 or Eq. equation 3. Given the $(\ell-1)$-st layer hidden representation matrix $\mathbf{Z}_{\mathcal{G}}^{(\ell-1)}$, we define $\mathbf{Z}_{\mathcal{G}}^{(\ell)}$ columnwise by

$$\mathbf{z}_{\mathcal{G},j}^{(\ell)} = \sigma\left(\sum_{i=1}^{F_{\ell-1}} \mathbf{Y}_{ij}^{(\ell)}\mathbf{z}_{\mathcal{G},i}^{(\ell-1)} + \mathbf{b}_j^{(\ell)}\right), \tag{4}$$

where $\mathbf{b}_j^{(\ell)}$ is a bias vector with equal real and imaginary parts, $\mathrm{Real}(\mathbf{b}_j^{(\ell)}) = \mathrm{Imag}(\mathbf{b}_j^{(\ell)})$. In matrix form, we write $\mathbf{H}_{\mathcal{G}}^{(\ell)} = \mathbf{Q}^{(\ell)}\left(\mathbf{H}_{\mathcal{G}}^{(\ell-1)}\right)$, where $\mathbf{Q}^{(\ell)}$ is a hidden layer of the form Eq. equation 4. In our experiments, we utilize convolutions of the form equation 3 with $\mathbf{L} = \mathbf{L}_N^{(q)}$ and set $K = 1$, in which case we obtain

$$\mathbf{Z}_{\mathcal{G}}^{(\ell)} = \sigma\left(\mathbf{Z}_{\mathcal{G}}^{(\ell-1)}\mathbf{W}_{\mathrm{self}}^{(\ell)} + \widetilde{\mathbf{L}}_N^{(q)}\mathbf{Z}_{\mathcal{G}}^{(\ell-1)}\mathbf{W}_{\mathrm{neigh}}^{(\ell)} + \mathbf{B}^{(\ell)}\right), \tag{5}$$

where $\mathbf{W}_{\mathrm{self}}^{(\ell)}$ and $\mathbf{W}_{\mathrm{neigh}}^{(\ell)}$ are learned weight matrices corresponding to the filter weights of different channels and $\mathbf{B}^{(\ell)} = (\mathbf{b}_1^{(\ell)}, \ldots, \mathbf{b}_{F_\ell}^{(\ell)})$.

## 3.2 Persistent Homology

PH is a subfield in computational topology, where the main goal is to detect, track, and encode the evolution of shape patterns in the observed object along various user-selected geometric dimensions (Edelsbrunner et al., 2000). These shape patterns represent topological properties such as connected components, loops, and, in general, $n$-dimensional "holes", that is, the characteristics of the graph $\mathcal{G}$ that remain preserved at different resolutions under continuous transformations. By employing such a multi-resolution approach, PH addresses the intrinsic limitations of classical homology and allows for retrieving the latent shape properties of $\mathcal{G}$ which may play an essential role in a given learning task. The key approach here is to select some suitable scale parameters $\nu$ and then to study changes in the shape of $\mathcal{G}$ that occur as $\mathcal{G}$ evolves concerning $\nu$. That is, we no longer study $\mathcal{G}$ as a single object but as a *filtration* $\mathcal{G}_{\nu_1} \subseteq \ldots \subseteq \mathcal{G}_{\nu_n} = \mathcal{G}$, induced by monotonic changes of $\nu$. To ensure that the process of pattern selection and counting is objective and efficient, we build an abstract simplicial complex $\mathscr{K}(\mathcal{G}_{\nu_j})$ on each $\mathcal{G}_{\nu_j}$, which results in filtration of complexes $\mathscr{K}(\mathcal{G}_{\nu_1}) \subseteq \ldots \subseteq \mathscr{K}(\mathcal{G}_{\nu_n})$. Note that, the abstract simplicial complex is a combinatorial structure specifying the adjacency of nodes, edges, triangles, and so on (Edelsbrunner & Harer, 2010). For example, for an edge-weighted graph $(\mathcal{V}, \mathcal{E}, w)$, with the edge-weight function $w : \mathcal{E} \to \mathbb{R}$, we can set $\mathcal{G}_{\leq \nu_j} = (\mathcal{V}, \mathcal{E}, w^{-1}((-\infty, v_j]))$ for each $\nu_j$, $j = 1, \ldots, n$, yielding the induced sublevel edge-weighted filtration. Similarly, we can consider a function on a node set $\mathcal{V}$, for example, node degree, which results in a sequence of induced subgraphs of $\mathcal{G}$ with a maximal degree of $\nu_j$ for each $j = 1, \ldots, n$ and the associated degree sublevel set filtration. We can then record scales $b_i$ (birth) and $d_i$ (death) at which each topological feature first and last appear in the sublevel filtration $\mathcal{G}_{\nu_1} \subseteq \mathcal{G}_{\nu_2} \subseteq \mathcal{G}_{\nu_3} \ldots \subseteq \mathcal{G}_{\nu_n}$. Figure 2 shows examples of degree-based filtration and power filtration.

Although the inherent nature of PH appears as a perfect fit to capture the topological characteristics of the graph, computational complexity remains the major roadblock on the way of wider adoption of PH in practice. For example, for 0-dimensional PH, the currently best available algorithm to compute PH has the complexity of $\mathcal{O} = (m\alpha(m))$, where $m$ denotes the number of simplices and $\alpha(\cdot)$ denotes inverse of the Ackermann function. One intuitive idea to address this fundamental problem is to use somehow only a subset of the available nodes when computing PH. *However, can we do so, without sacrificing the topological information?* The answer to this question is *positive* if we invoke the notion of a Dowker complex on graphs. Dowker complex

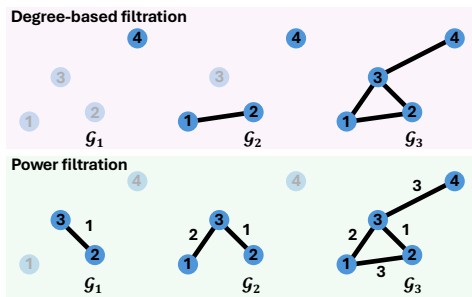

Figure 2: Degree-based filtration (upper) and power filtration (bottom).

belongs to the family of weaker complexes which also includes, for example, witness complex (Ghrist, 2014; Chazal et al., 2014; Chowdhury & Mémoli, 2018), and also may be viewed as the witness complex counterpart on graphs (for more discussion on similarities and differences of witness and Dowker complexes see Aksoy et al. (2023)). The ultimate idea is to assess the shape of the graph based only on a substantially smaller subset of nodes, called *landmarks*, while using all other remaining nodes as *witnesses* which dictate appearances of simplices in the Dowker complex.

**Definition 1. *Dowker Complex.*** *For a graph $\mathcal{G} = (\mathcal{V}, \mathcal{E})$, the Dowker Complex $D(L, W)$ is a simplicial complex constructed from the landmark set $L$ and the witness set $W$ as follows:*

$$D(\mathcal{G}) = \left\{\sigma \subseteq L \mid \exists w \in W \ such \ that \quad \forall l \in \sigma, \ d(l, w) \leq \epsilon\right\},$$

*where $\varepsilon$ represents the maximum allowable distance between landmark nodes and witness nodes. Each simplex $\sigma$ in $D(L, W)$ corresponds to a subset of landmark nodes. A $k$-simplex is included in $D(L, W)$ if there exists at least one witness node $w \in W$ that is within distance $\varepsilon$ from every landmark node in $\sigma$.*

To select the suitable set $L$ of landmark nodes for the Dowker complex, we leverage the $\varepsilon$-nets algorithm (De Silva & Carlsson, 2004; Arafat et al., 2020; 2025), which ensures computational efficiency without loss of topological information. Given the Dowker complex of the graph $D(\mathcal{G})$, we compute the Dowker complex-based persistence diagram (i.e., DC-PD) as follows $DC_{Dg} = \Xi(D(\mathcal{G}))$ where $\Xi$ denotes the function that computes Dowker persistence.

In this paper, to encode the above topological information presented in a DC-PD $DC_{Dg}$ into the embedding function, we use its vectorized representation, i.e., persistence image (PI) (Adams et al., 2017). The PI is a finite-dimensional vector representation obtained through a weighted kernel density function and can be computed in the following two steps (see more details in Definition 2). First, we map the DC-PD $DC_{Dg}$ to an integrable function $\varrho_{DC_{Dg}} : \mathbb{R}^2 \mapsto \mathbb{R}^2$, which is referred to as a persistence surface. The persistence surface $\varrho_{DC_{Dg}}$ is constructed by summing weighted Gaussian kernels centered at each point in $DC_{Dg}$. Second, we integrate the persistence surface $\varrho_{DC_{Dg}}$ over each grid box to obtain the value of the Dowker complex-based PI (i.e., $DC_{PI}$) (see its definition 2 in Appendix A).

For a given DC-PD vectorization (e.g., the differentiable distribution-based transformation described above), stability is a fundamental property from a statistical perspective. That is, stability requires that small perturbations in the input DC-PD lead to only small changes in its vectorized representation. To formalize this notion, distances between DC-PDs are measured using the Wasserstein distance, which provides a principled metric for comparing persistence diagrams. A vectorization function is said to be stable if the distance between vectorized representations is bounded by a constant multiple of the Wasserstein distance between the corresponding DC-PDs. This guarantees robustness of the learned topological features to small perturbations in the underlying graph structure. Formal definitions of the Wasserstein distance and the associated stability results are provided in Appendix A. Figure 3 displays different network structures and their corresponding Dowker complex-based persistence images.

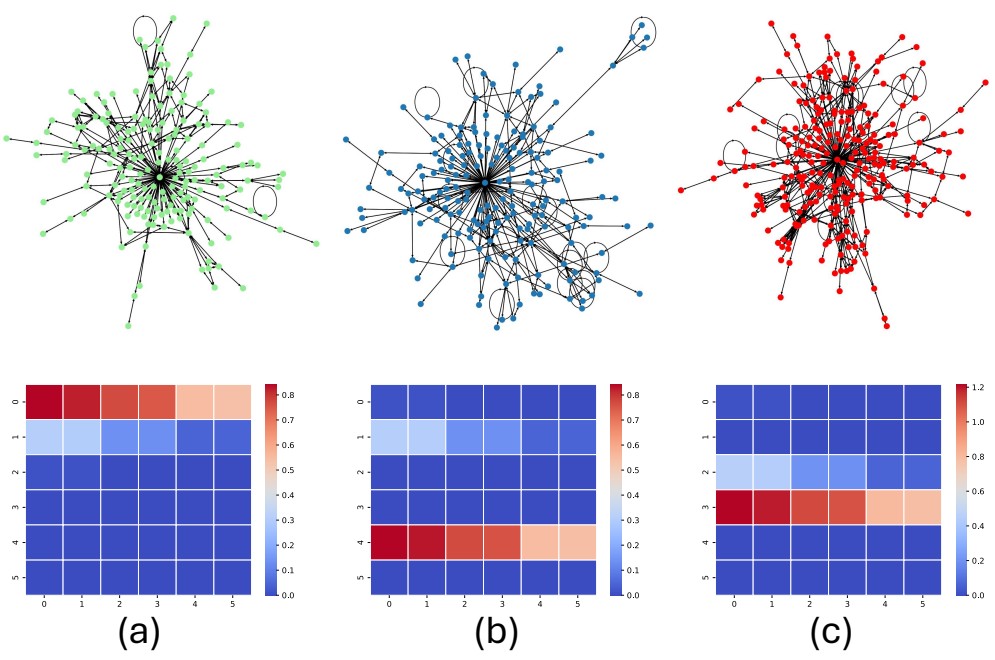

Figure 3: Illustration of network structures and their corresponding Dowker complex-based persistence images for (a) Texas, (b) Cornell, and (c) Wisconsin respectively.

### 3.3 Topological Representation Learning

To capture the underlying topological features of the graph $\mathcal{G}$, we employ $\mathcal{K}$ filtration functions: $f_i : \mathcal{V} \mapsto \mathbb{R}$ for $i = \{1, \ldots, \mathcal{K}\}$. Each filtration function $f_i$ gradually reveals one specific topological structure at different levels of connectivity, e.g., degree centrality score, betweenness centrality score, closeness centrality score, and other node centrality measurements. With each filtration function $f_i$, we construct a set of two persistence images of resolution $P \times P$ using tools in PH analysis (since we focus on both 0- and 1-dimensional topological features). Combining two persistence images of resolution $P \times P$ from $\mathcal{K}$ different filtration functions, we construct a *multi-view* topological representation, i.e., the set of Dowker complex-based PIs $[DC_{PI_1}^{(0)}, DC_{PI_1}^{(1)}, DC_{PI_2}^{(0)}, DC_{PI_2}^{(1)}, \ldots, DC_{PI_\mathcal{K}}^{(0)}, DC_{PI_\mathcal{K}}^{(1)}]$ with the dimension $\mathcal{K} \times 2 \times P \times P$. We design a topological convolutional layer $\Phi(\cdot)$ to (i) jointly extract and learn the latent topological features and (ii) leverage and preserve the multi-dimensional graph structural information. Firstly, hidden representations of the set of PIs are achieved through a combination of a CNN-based model and global pooling, which can be defined as

$$\mathbf{Z}_\mathcal{T} = \Phi([\mathbf{X}, DC_{PI_1}^{(0)}, DC_{PI_1}^{(1)}, DC_{PI_2}^{(0)}, DC_{PI_2}^{(1)}, \ldots, DC_{PI_\mathcal{K}}^{(0)}, DC_{PI_\mathcal{K}}^{(1)}])), \tag{6}$$

where $\Phi(\cdot)$ embeds Dowker complex-based PIs to a high-dimensional space, and $[\cdots, \cdots]$ denotes the concatenation operation. This step is crucial for graph representation learning as the resulting topological embedding contains rich topological information encoded in the graph and can be integrated into any graph prompting functions. Based on $\mathbf{Z}_\mathcal{G}$ and $\mathbf{Z}_\mathcal{T}$, we utilize the aggregation and combination operation to obtain the node representations which is defined as follows

$$\mathbf{Z} = \text{AGG-COMB}(\mathbf{Z}_\mathcal{G}, \mathbf{Z}_\mathcal{T}), \tag{7}$$

where AGG-COMB operation collects the node and topology embeddings, and combines them using sum, mean, or max functions to generate the joint embedding for nodes. In this work, we employ the link prediction task for pre-training, i.e., deciding if nodes are connected which is generated as $\min_{\xi, \zeta} \sum_{u,v} \mathcal{L}^{pre}(f_\zeta(\mathbf{z}_u, \mathbf{z}_v); g(u, v))$, where $f_\zeta$ is the projection head with trainable parameters $\zeta$ (we adopt the Multi-Layer Perceptrons (MLP)) which computes the similarity between node embeddings, and $g(u, v)$ is the label of node pair (i.e., if any pair of nodes are connected or not) based on the adjacency matrix $\mathbf{A}$.

### 3.4 Prompting for Downstream Tasks

Most existing graph prompting methods construct prompts purely from node-level attributes, where a prompting function $f_{\text{prompt}}(\cdot)$ maps a node's standalone feature vector into a structured token. While effective in undirected or feature-rich settings, such approaches are often insufficient for directed graphs where label semantics are governed by asymmetric flows and higher-order structural patterns that cannot be fully captured by node features alone. To address this limitation, in this paper, we introduce a topology-empowered prompting strategy that injects the joint directional and topological representations learned during pre-training directly into the downstream prompt tokens. Specifically, after fusing the magnetic Laplacian–based structural embedding $\mathbf{Z}_{\mathcal{G}}$ and the Dowker complex–based topological embedding $\mathbf{Z}_{\mathcal{T}}$ into a joint node representation $\mathbf{Z}$ (see Eq. 7), we treat the resulting node embedding $\mathbf{z}_u$ as a topology-assisted structure token for each node $u \in V_{\text{prompt}} \subseteq V$, denoted by

$$\boldsymbol{\xi}_u^{\text{TG}} = \mathbf{z}_u. \tag{8}$$

That is, the obtained representation encodes both local directional connectivity and higher-order topological signatures which enables the prompt to reflect intrinsic graph organization rather than isolated node attributes. Following the GPPT-based prompting paradigm (Sun et al., 2022), we reformulate downstream node classification as a compatibility prediction task between a node and a class-specific task token. Let $\{y_1, \ldots, y_C\}$ denote the set of class labels (total $C$ classes), and associate each class $y_c$ with a learnable task token $T_{\text{task}}(y_c)$. The final prompt token for node $u$ under class $c$ is constructed as

$$\xi_{\text{task},u,c}^{\text{TG}} = f_{\text{prompt}}^{\text{TG}}(u, c) = \left[\, T_{\text{task}}(y_c),\ \xi_u^{\text{TG}} \,\right], \tag{9}$$

where $[\cdot, \cdot]$ denotes concatenation. Intuitively, the task token $T_{\text{task}}(y_c)$ acts as a class-specific query, while the topology-assisted structure token $\xi_u^{\text{TG}}$ provides a topology-guided representation of node $u$, allowing the model to retrieve task-relevant structural knowledge encoded during pre-training. Given a pre-trained projection head $f_\phi^{\text{pre}}(\cdot)$, we compute a compatibility score between node $u$ and class $c$ as

$$s(u, c) = f_\phi^{\text{pre}}\left(\xi_{\text{task},u,c}^{\text{TG}}\right). \tag{10}$$

The downstream objective is then formulated as a cross-entropy loss over the prompted nodes

$$\min_{\theta, \phi, E^1, \ldots, E^M} \sum_{(u, y_c)} \mathcal{L}^{\text{pre}}\Big(s(u, c); \mathbb{I}(T_{task}(y_c), u)\Big), \quad \text{s.t.} \quad \theta^{\text{init}} = \theta^{\text{pre}}, \phi^{\text{init}} = \phi^{\text{pre}}. \tag{11}$$

where $\mathbb{I}(\cdot)$ denotes the ground-truth label associated with a node pair, $E^1, \ldots, E^M$ denote $M$ sets of learnable prompt embedding matrices (i.e., following Sun et al. (2022), we first adopt METIS (Karypis & Kumar, 1998) to pre-process and split nodes into $M$ non-overlapped clusters, as varying task tokens in different clusters may provide enhanced task embeddings), where each $E^m = [e_1^m, \ldots, e_C^m]^\top \in \mathbb{R}^{C \times d}$ stores the class-specific prompt tokens $e_c^m$ associated with label $y_c$ (these embeddings are optimized during downstream prompt tuning), and $\mathcal{L}^{\text{pre}}(\cdot; \cdot)$ denotes the cross-entropy loss function. This optimization strategy preserves the pre-trained encoder and projection parameters while enabling task adaptation through prompt tuning, i.e., reducing task mismatch and improving generalization in few-shot settings. In contrast to conventional graph prompting approaches (which rely solely on node attributes), our proposed method prompts on topology-assisted representations that integrate directional structural cues and higher-order connectivity patterns. This design allows downstream tasks to directly leverage causally relevant graph structure learned during pre-training, leading to more robust and transferable performance on directed graph classification tasks.

## 4 Experiments

### 4.1 Datasets

In this study, we evaluate TopoDIG on 10 real-word and synthetic datasets. For 5 real-world directed graphs, we use (i) Texas, Wisconsin, and Cornell which belong to the WebKB collection (Pei et al., 2020) and these

datasets capture hyperlink connections between webpages from computer science departments at different universities. In each network, nodes represent webpages, edges indicate hyperlinks, and features are based on a bag-of-words model. The webpages are manually classified into 5 categories such as student, project, course, staff, and faculty; and (ii) ogbn-arxiv and ogbn-Papers100M from the Open Graph Benchmark (OGB) (Hu et al., 2020) which are large-scale citation networks representing paper–paper citation relationships. In these datasets, nodes denote scientific papers, directed edges denote citation links, and node features are derived from paper abstracts. The node labels indicate subject areas, with 40 classes in ogbn-arxiv and 172 classes in ogbn-Papers100M. For synthetic data, we use Directed Stochastic Block Models (DSBMs) introduced in He et al. (2022b), which generate directed graphs with clusters defined by network flows between groups/blocks. Each block represents a cluster in the directed graph, where clustering is a partition of the set of nodes into $K$ disjoint sets (clusters) $\mathcal{V} = \mathcal{C}_0 \cup \mathcal{C}_1 \cup \cdots \cup \mathcal{C}_{K-1}$ (ideally, $K \geq 2$). In Appendix A, we provide detailed descriptions of synthetic graph generation, and Tables 1 and 2 show dataset statistics of real-world and synthetic graph datasets.

Table 1: Overview of the real-world datasets.

| Dataset | Cornell | Texas | Wisconsin | ogbn-arxiv | ogbn-Papers100M |
|---|---|---|---|---|---|
| # Nodes | 183 | 183 | 251 | 169,343 | 111,059,956 |
| # Edges | 295 | 309 | 499 | 1,166,243 | 1,615,685,872 |
| # Features | 1,703 | 1,703 | 1,703 | 128 | 128 |
| # Classes | 5 | 5 | 5 | 40 | 172 |

Table 2: Overview of the synthetic datasets.

| Dataset | Cycle (4, 0.1) | Cycle (4, 0.01) | Complete (5, 0.01) | Star (5, 0.02) | Path (5, 0.02) |
|---|---|---|---|---|---|
| # Nodes | 1,000 | 1,000 | 1,000 | 1,000 | 1,000 |
| # Edges | 50,052 | 5,021 | 4,900 | 10,309 | 9,985 |
| # Features | 8 | 8 | 8 | 8 | 8 |
| # Classes | 4 | 5 | 5 | 5 | 5 |

Table 3: Evaluation results for node classification on Texas, Cornell, and Wisconsin, reporting the mean plus/minus one standard deviation over 5 runs. Best results are in **bold**.

| Method | Texas | Cornell | Wisconsin |
|---|---|---|---|
| GCN | $57.81 \pm 0.78$ | $53.91 \pm 1.98$ | $47.54 \pm 3.10$ |
| GAT | $62.50 \pm 0.78$ | $56.56 \pm 4.09$ | $49.49 \pm 3.82$ |
| GIN | $60.16 \pm 2.34$ | $51.41 \pm 0.58$ | $41.14 \pm 1.95$ |
| GraphTransformer | $62.50 \pm 3.12$ | $56.87 \pm 3.18$ | $\mathbf{58.17 \pm 1.93}$ |
| DiGCN | $48.44 \pm 9.94$ | $44.53 \pm 4.50$ | $47.66 \pm 3.50$ |
| MagNet | $58.98 \pm 2.73$ | $60.00 \pm 1.67$ | $55.54 \pm 3.85$ |
| GCN+CNA | $59.73 \pm 6.01$ | $60.12 \pm 1.55$ | $56.79 \pm 3.61$ |
| AdapterGNN | $57.47 \pm 4.31$ | $59.61 \pm 1.79$ | $57.80 \pm 1.52$ |
| **TopoDIG (Ours)** | $\mathbf{64.84 \pm 2.34}$ | $\mathbf{61.56 \pm 2.49}$ | $57.94 \pm 2.86$ |

## 4.2 Baselines and Experiment Setups

In our experiments, we compare our approach with representative graph neural networks, i.e., Graph Convolutional Networks (GCN) (Kipf & Welling, 2017), Graph Isomorphism Network (GIN) (Xu et al., 2019), Graph Attention Networks (GAT) (Veličković et al., 2018), and GCN+Cluster+Normalize+Activate (CNA) modules (Skryagin et al., 2024), a transfer learning method, i.e., AdapterGNN (Li et al., 2024), and a graph transformer model (i.e., GraphTransformer) (Shi et al., 2021). For directed graphs specifically, we compare TopoDIG with DiGCN (Tong et al., 2020a) and MagNet (Zhang et al., 2021b). We train all

Table 4: Evaluation results for 3-shot node classification (in %) on ogbn-arxiv.

| Method | ogbn-arxiv |
|---|---|
| GPPT (Sun et al., 2022) | $23.35 \pm 0.70$ |
| All-in-one (Sun et al., 2023a) | $16.79 \pm 6.81$ |
| GraphPrompt (Liu et al., 2023) | $82.07 \pm 3.19$ |
| GPF (Fang et al., 2023) | $77.50 \pm 7.23$ |
| **TopoDIG (Ours)** | $\mathbf{86.78 \pm 2.20}$ |

Table 5: Results (in %) of fine-tuning the pretrained GNN with 1% and 5% labeled training data on ogbn-Papers100M.

| Method | Label ratio 1% | Label ratio 5% |
|---|---|---|
| CCA-SSG (Zhang et al., 2021a) | $55.68 \pm 0.24$ | $59.78 \pm 0.08$ |
| GRACE (Zhu et al., 2020) | $55.45 \pm 0.23$ | $59.38 \pm 0.15$ |
| BGRL (Thakoor et al., 2022) | $55.12 \pm 0.23$ | $60.40 \pm 0.54$ |
| GraphMAE (Hou et al., 2022) | $58.29 \pm 0.15$ | $62.00 \pm 0.12$ |
| GraphMAE2 (Hou et al., 2023) | $58.69 \pm 0.38$ | $62.87 \pm 0.64$ |
| **TopoDIG (Ours)** | $\mathbf{59.42 \pm 0.41}$ | $\mathbf{64.05 \pm 0.59}$ |

Table 6: Evaluation results for node classification on synthetic datasets, reporting the mean plus/minus one standard deviation over 5 runs. Best results are in **bold**.

| Method | Cycle (4, 0.1) | Cycle (4, 0.01) | Complete (5, 0.01) | Star (5, 0.02) | Path (5, 0.02) |
|---|---|---|---|---|---|
| GCN | $26.97 \pm 1.48$ | $31.94 \pm 1.25$ | $36.91 \pm 0.98$ | $37.20 \pm 1.87$ | $39.26 \pm 1.74$ |
| GAT | $24.11 \pm 1.07$ | $31.40 \pm 1.47$ | $35.29 \pm 1.59$ | $30.66 \pm 1.20$ | $31.46 \pm 1.61$ |
| GIN | $27.66 \pm 5.73$ | $38.00 \pm 1.12$ | $43.29 \pm 0.52$ | $25.71 \pm 1.91$ | $23.66 \pm 1.15$ |
| GraphTransformer | $38.86 \pm 2.16$ | $41.31 \pm 1.56$ | $45.57 \pm 2.08$ | $37.03 \pm 1.51$ | $43.71 \pm 2.28$ |
| DiGCN | $24.60 \pm 1.57$ | $27.11 \pm 1.01$ | $27.49 \pm 1.66$ | $20.94 \pm 0.62$ | $22.51 \pm 1.62$ |
| MagNet | $26.69 \pm 8.35$ | $39.26 \pm 1.48$ | $46.89 \pm 0.735$ | $35.97 \pm 4.02$ | $43.26 \pm 2.06$ |
| GCN+CNA | $38.74 \pm 4.98$ | $46.65 \pm 2.81$ | $44.01 \pm 2.30$ | $36.25 \pm 4.35$ | $41.65 \pm 1.53$ |
| AdapterGNN | $38.00 \pm 4.01$ | $35.15 \pm 2.57$ | $40.74 \pm 3.06$ | $38.30 \pm 1.13$ | $39.10 \pm 1.02$ |
| **TopoDIG (Ours)** | $\mathbf{50.26 \pm 4.52}$ | $\mathbf{71.43 \pm 12.39}$ | $\mathbf{47.31 \pm 1.16}$ | $\mathbf{43.11 \pm 3.10}$ | $\mathbf{43.97 \pm 3.91}$ |

Table 7: Ablation experiment results, reporting the mean plus/minus one standard deviation over 5 runs. Best results are in **bold**.

| Method | Texas | Cornell | Wisconsin |
|---|---|---|---|
| w/o MLGC | $60.09 \pm 5.74$ | $59.56 \pm 1.36$ | $56.14 \pm 3.80$ |
| w/o Topo | $58.32 \pm 4.18$ | $60.25 \pm 2.02$ | $57.83 \pm 1.37$ |
| **TopoDIG (Ours)** | $\mathbf{64.84 \pm 2.34}$ | $\mathbf{61.56 \pm 2.49}$ | $\mathbf{57.94 \pm 2.86}$ |

"pre-train and prompt" models on 4 NVIDIA RTX A5000 GPU cards with 24GB memory, and report the mean accuracy (in %) and standard deviation over 5 runs with different random seeds. In the pre-training phase, we use cross-entropy loss on positive (i.e., adjacency matrix) and negative (via random sampling) connected edges for link prediction. We save the best-performed pre-trained model parameter within 1000 epochs using an early stop strategy for downstream prompting tasks. Since this paper mainly focuses on the pretraining strategy, we leverage a prompting strategy proposed in Sun et al. (2022) for downstream few-shot node classification tasks. To provide fairness comparisons, we set the seed to 42 and tune each model with learning rate in $\{5e^{-3}, 5e^{-4}, 5e^{-5}, 5e^{-6}\}$, hidden dimension in $\{128, 256, 512\}$, number of GNN layers in $\{2, 3\}$. Additionally, "charge parameter" q used in MagNet and TopoDIG tune within

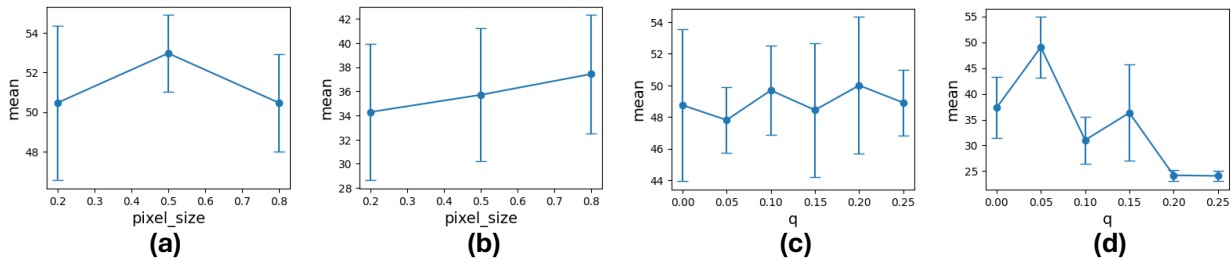

Figure 4: (a) and (b): Sensitivity study of the pixel size of the persistence image on Cornell and Cycle (5, 0.02). (c) and (d): Sensitivity study of the parameter $q$ on Cornell and Cycle (4, 0.1). All plots show the mean (dot) and variance (error bar) on 5 runs.

Table 8: Graph prompt function performance comparison (Accuracy±std).

| Method | Texas | Cornell | Wisconsin |
|---|---|---|---|
| TopoDIG with GPPT (ours) | $64.84 \pm 2.34$ | $\mathbf{61.56 \pm 2.49}$ | $57.94 \pm 2.86$ |
| TopoDIG with GPrompt | $63.12 \pm 2.58$ | $59.91 \pm 2.41$ | $55.47 \pm 2.77$ |
| TopoDIG with GPF | $\mathbf{65.27 \pm 2.41}$ | $60.38 \pm 2.57$ | $\mathbf{58.42 \pm 2.73}$ |

$\{0.05, 0.1, 0.15, 0.2, 0.25\}$, and pixel size for generating the topological features tuned within $\{0.2, 0.5, 0.8\}$. Code can be found at `https://github.com/ChenguangYang96UCR/TopoDIG`.

## 4.3 Results

We provide few-shot node classification results within the pre-train and prompt paradigm in Tables 3, 4, 5, and 6, while seeing that TopoDIG performs well across all datasets. As shown in Table 3, our proposed TopoDIG outperforms all baselines on Texas and Cornell except for Wisconsin (which may be due to Graph-Transformer utilizes global attention to enhance information aggregation from sparse data and TopoDIG still achieves the runner-up result on Wisconsin). Based on the results in Tables 4 and 5, our method TopoDIG consistently and substantially outperforms all competing baselines across both few-shot and low-label regimes. On the 3-shot node classification task on ogbn-arxiv (see Table 4), TopoDIG achieves the highest accuracy, surpassing the strongest baseline Gprompt by a clear margin, while dramatically outperforming earlier prompting-based and fine-tuning approaches. This demonstrates the effectiveness of TopoDIG in extremely label-scarce settings on large-scale directed citation graphs. For fine-tuning with limited labeled data (see Table 5), TopoDIG again delivers the best performance under both 1% and 5% label ratios, respectively. Compared to the strongest self-supervised baseline, GraphMAE2, TopoDIG yields consistent gains while maintaining comparable variance, highlighting its superior representation quality and robustness. Overall, these results confirm that incorporating topology-aware representations enables TopoDIG to more effectively leverage structural information, leading to superior performance on large-scale directed graphs. In Table 6, we use 5 synthetic graphs with different structures. While setting the total number of nodes per graph invariant to 1000, and we observe that TopoDIG outperforms all baselines across various graph densities and connectivity patterns with a relative average gain of 25.84% overall 5 synthetic datasets. Furthermore, we provide a detailed computational complexity analysis in Appendix B.

## 4.4 Ablation Studies

To evaluate the effectiveness of each module in TopoDIG, we compare it with 2 model variants. Specifically, "w/o SGNN" and "w/o Topo" represent methods without using MLGC module and topological representation learning module (Topo). Table 7 shows results on Texas, Cornell, and Wisconsin. It is shown that TopoDIG consistently outperforms 2 variants on all 3 datasets, thereby demonstrating the effectiveness of each module in TopoDIG. For the ablation scope, we have conducted additional experiments, i.e., except GPPT, we utilize GraphPrompt (Liu et al., 2023) and GPF (Fang et al., 2023) for directed graph learning on

Texas, Cornell, and Wisconsin data. As shown in Table 8, we observe that (1) TopoDIG with GPPT always outperform TopoDIG with Gprompt; (2) TopoDIG with GPF achieves the highest accuracy on Texas and Wisconsin data, however, it achieves a slightly lower accuracy compared to TopoDIG with GPPT on Cornell; (3) compared to other baselines (shown in Table 3), TopoDIG with different prompt functions achieves state-of-the-art performances.

### 4.5 Sensitivity Analysis

We conduct a sensitivity analysis on pixel size in generating topological features, where increasing the pixel count results in smaller persistence image sizes. As shown in Figure 4 (a & b), different datasets achieve optimal performance with different pixel sizes. For downstream node classification tasks, it is important to ensure that the connectivity signal does not dominate the node features themselves. Therefore, the size of the input node feature plays a crucial role when selecting the pixel size. In the cases of Cornell and Cycle(5, 0.02), which have input feature sizes of 1,703 and 8, respectively, the best performance is observed at pixel sizes of 0.5 and 0.8. We also conduct sensitivity analysis of the parameter $q$. Figure 4 (c & d) shows results of sensitivity analysis on Cornell and Cycle (4, 0.1) datasets. We find that, on Cornell data, the performance remains relative stable across different values of $q$ which suggests that our model is not highly sensitive to the $q$ on this data which is a sparse graph. However, on Cycle (4, 0.1), the performance fluctuates significantly with different values of $q$ (especially, we can observe a sharp decline at $q = 0.05$ and $q = 0.15$) which indicates that the parameter $q$ selection is crucial for dense graph analysis.

## 5 Conclusion

In conclusion, this work introduces the Topology-Guided Directed Graph Pre-training and Prompt Learning (TopoDIG), a pioneering framework designed to the graph prompting study on directed graphs with applying magnetic Laplacian and persistent homology. Through a novel integration of magnetic Laplacian-based graph convolutional networks module and Dowker complex-based topological representation learning module, TopoDIG effectively captures directional and local topological information of directed graphs. Our experiments demonstrate our TopoDIG's superior performance on various real-world and synthetic datasets. In the future, we will extend our approach to dynamic directed graph learning tasks.

## Acknowledgments

Y.H. is supported by an Nvidia Academic Grant award and a Jetstream2 NAIRR AI Fellowship. Y.C. was supported in part by the NSF DMS-2523484, DMS-2420959, and UCR SoCal OASIS Funding Award. Any opinions, findings, and conclusions or recommendations expressed in this document are those of the author(s) and do not necessarily reflect the views of the National Science Foundation or other funding parties.

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

# A Datasets and Visualization

Here we provide more details of the DSBM model. A DSBM model is defined by the number of clusters, $K$, and edge probabilities, with edges being independently assigned given cluster memberships. In our experiments, the DSBM is characterized by a meta-graph adjacency matrix $\mathbf{F} = (\mathbf{F}_{k,l})_{k,l=0,\ldots,K-1}$, its filled version $\tilde{\mathbf{F}} = (\tilde{\mathbf{F}}_{k,l})_{k,l=0,\ldots,K-1}$, and a noise level $\eta \leq 0.5$. The matrix $\mathbf{F}$ is derived from a meta-graph structure $\mathcal{M}$. Here we do not have an ambient background as in He et al. (2022b). We follow He et al. (2022b) to define four structures of $\mathbf{F}$ without any ambient nodes, where $\mathbb{1}$ denotes the indicator function. In this study, we choose the number of clusters, $K$, the meta-graph structure, and the edge probability $p$. We set the number of nodes $n$ to be 1000 and the (approximate) ratio, $\rho$, between the largest and the smallest cluster size, to be 1, and the direction flip parameter $\eta = 0.05$.

- "*Cycle*": $\mathbf{F}_{k,l} = (1-\eta)\mathbb{1}(l = ((k+1) \mod K)) + \eta\mathbb{1}(l = ((k-1) \mod K)) + \frac{1}{2}\mathbb{1}(l = k)$.

- "*Path*": $\mathbf{F}_{k,l} = (1-\eta)\mathbb{1}(l = k+1) + \eta\mathbb{1}(l = k-1) + \frac{1}{2}\mathbb{1}(l = k)$.

- "*Complete*": assign diagonal entries $\frac{1}{2}$. For each pair $(k,l)$ with $k < l$, let $\mathbf{F}_{k,l}$ be $\eta$ and $1-\eta$ with equal probability, then assign $\mathbf{F}_{l,k} = 1 - \mathbf{F}_{k,l}$.

- "*Star*": select the center node as $\omega = \lfloor \frac{K-1}{2} \rfloor$ and set $\mathbf{F}_{k,l} = (1-\eta)\mathbb{1}(k = \omega, l \text{ odd}) + \eta\mathbb{1}(k = \omega, l \text{ even}) + (1-\eta)\mathbb{1}(l = \omega, k \text{ odd}) + \eta\mathbb{1}(l = \omega, l \text{ even})$.

For synthetic graph generation, in our experiments, we choose the number of clusters, $K$, the meta-graph structure, and the edge probability $p$. We set the number of nodes $n$ to be 1000 and the (approximate) ratio, $\rho$, between the largest and the smallest cluster size, to be 1, and the direction flip parameter $\eta = 0.05$. Our DSBM, which we denote by $\mathcal{M}(K, p, \rho)$ is built according to He et al. (2022b). Due to $\rho$ is set to 1, we simplify the notation to $\mathcal{M}(K, p)$ and $\mathcal{M}$ represents the structures, such as, "*Cycle*", "*Path*", "*Complete*", and "*Star*". For each node $v_i \in \mathcal{C}_k$, and each node $v_j \in \mathcal{C}_l$, independently sample an edge from node $v_i$ to node $v_j$ with probability $p \cdot \mathbf{F}_{k,l}$. Based on He et al. (2022b), we use the $\mathcal{O}^{\text{sort}}_{\text{vol\_sum}}$ term with the naive pair selection method to compute an imbalance score for the ground-truth labels for our datasets, in order to explain the level of flow imbalance related to directionality. Roughly speaking, the larger this imbalance score, the more directed this network is, with respect to its class labels.

Wasserstein distance (or matching distance) is defined as follows. Let $DC_{Dg}(\mathcal{G}^+)$ and $DC_{Dg}(\mathcal{G}^-)$ be Dowker complex-based persistence diagrams of two graphs $\mathcal{G}^+$ and $\mathcal{G}^-$ (We omit the dimensions in DC-PDs). Let $DC_{Dg}(\mathcal{G}^+) = \{q_j^+\} \cup \Delta^+$ and $DC_{Dg}(\mathcal{G}^-) = \{q_l^-\} \cup \Delta^-$ where $\Delta^\pm$ represents the diagonal (representing trivial cycles) with infinite multiplicity. Here, $q_j^+ = (b_j^+, d_j^+) \in DC_{Dg}(\mathcal{G}^+)$ represents the birth and death times of a $k$-hole $\sigma_j$. Let $\phi : DC_{Dg}^k(\mathcal{G}^+) \to DC_{Dg}^k(\mathcal{G}^-)$ represent a bijection (matching). With the existence of the diagonal $\Delta^\pm$ on both sides, we make sure of the existence of these bijections even if the cardinalities $|\{q_j^+\}|$ and $|\{q_l^-\}|$ are different. Then, $p^{th}$ Wasserstein distance $f_{\mathcal{W}_p}$ defined as $f_{\mathcal{W}_p}(DC_{Dg}(\mathcal{G}^+), DC_{Dg}(\mathcal{G}^-)) = \min_\phi(\sum_j \|q_j^+ - \phi(q_j^+)\|_\infty^p)^{\frac{1}{p}}$, where $p \in \mathbb{Z}^+$. Here, the bottleneck distance is $f_{\mathcal{W}_\infty}(DC_{Dg}(\mathcal{G}^+), DC_{Dg}(\mathcal{G}^-)) = \max_j \|q_j^+ - \phi(q_j^+)\|_\infty$. Then, function $\varphi$ is called *stable* if $\mathrm{d}(\varphi^+, \varphi^-) \leq C \cdot f_{\mathcal{W}_p}(DC_{Dg}(\mathcal{G}^+), DC_{Dg}(\mathcal{G}^-))$, where $\varphi^\pm$ is a vectorization of $DC_{Dg}(\mathcal{G}^\pm)$ and $\mathrm{d}(.,.)$ is a suitable metric in the space of vectorizations. Here, the constant $C > 0$ is independent of $\mathcal{G}^\pm$. This stability inequality interprets that as the changes in the vectorizations are bounded by the changes in DC-PDs.

**Definition 2** (Dower Complex-Based Persistence Image). *Let $g : \mathbb{R}^2 \mapsto \mathbb{R}$ be a non-negative weight function for the persistence plane $\mathbb{R}$. The value of each pixel $z \in \mathbb{R}^2$ is defined as $DC_{PI}(z) = \iint_z \sum_{\mu \in T(DC_{Dg})} \frac{g(\mu)}{2\pi\delta_x\delta_y} e^{-\left(\frac{(x-\mu_x)^2}{2\delta_x^2} + \frac{(y-\mu_y)^2}{2\delta_y^2}\right)} dy dx$, where $T(DC_{Dg})$ is the transformation of the $DC_{Dg}$ (i.e., for each $(x,y)$, $T(x,y) = (x, y-x)$), $\mu = (\mu_x, \mu_y) \in \mathbb{R}^2$, and $\delta_x$ and $\delta_y$ are the standard deviations of a differentiable probability distribution in the $x$ and $y$ directions, respectively.*

# B  Computational Complexity

The computational complexity of persistent homology (PH) for persistence diagram is $\mathcal{O}((n+m)^\omega)$ where $\omega = 2.3728596$. Dowker complex can reduce the complexity of computing 0- and 1-dimensional features to $\mathcal{O}((n'+m')^\omega)$ where $n'$ denotes the number of landmark nodes, and $m'$ is the number of edges in the subgraph induced by those landmarks. For the magnetic Laplacian module, the computational complexity is $O(|E| \cdot F_{\ell-1} + |V| \cdot F_{\ell-1}F_\ell)$ where $F_{\ell-1}$ denotes the embedding dimension of each node at the input of layer $\ell$ and $F_\ell$ denotes the embedding dimension of each node at the output of layer $\ell$. The complexity of CNN + pooling topological encoder is $O(\mathcal{K} \cdot P^2 \cdot C_{CNN})$ where $\mathcal{K}$ denotes the number of filtration functions, $P$ denotes the resolution size of the persistence image, and $C_{CNN}$ denotes the interaction between input and output channels in standard convolution operations. We also compare the running time (training time per epoch; along with the accuracy (%)) between our TopoDIG model and 5 runner-ups, i.e., DGCN, GCN+CNA, AdapterGNN, GPF (Fang et al., 2023), and GraphMAE2 (Hou et al., 2023). From Table 9, our TopoDIG consistently achieves competitive or faster training efficiency compared to all baselines on small-scale benchmarks (i.e., Texas, Cornell, Wisconsin), the large-scale graph (i.e., ogbn-arxiv), and synthetic settings (i.e., Cycle (4, 0.01) and Complete (5, 0.01)). Importantly, despite its favorable computational efficiency, TopoDIG also delivers consistently superior accuracy across all datasets (see Tables 3, 6, 4, 5), which indicates a more effective trade-off between representational expressiveness and computational cost. These results demonstrate that the proposed TopoDIG enables improved direct link prediction performance without incurring additional training overhead, and in many cases, and offers both accuracy and efficiency gains over existing methods. Furthermore, for the ogbn-arxiv data, as shown in Table 10, the topological preprocessing requires 5.0 minutes for Dowker complex construction and 1.5 minutes for persistence image generation, and is performed once per graph and reused across runs. Training takes 25.30 seconds per epoch and full-graph inference requires 2.6 seconds. During topological preprocessing (Dowker complex construction and persistence image computation), the peak CPU memory usage is around 35 GB. During model training, the GPU memory peak is around 9.0 GB. The resulting topological features are compact and reusable, which requires around 0.5 GB.

Table 9: Running time analysis (seconds per epoch).

| Model | Texas | Cornell | Wisconsin | ogbn-arxiv | Cycle (4, 0.01) | Complete (5, 0.01) |
|---|---|---|---|---|---|---|
| DGCN | 1.38 s | 0.50 s | 0.32 s | 34.43 s | 0.69 s | 0.58 s |
| GCN+CNA | 0.82 s | 0.29 s | 0.18 s | 21.60 s | 0.40 s | 0.35 s |
| AdapterGNN | 0.96 s | 0.34 s | 0.22 s | 24.80 s | 0.48 s | 0.42 s |
| GPF | 1.10 s | 0.40 s | 0.26 s | 28.90 s | 0.56 s | 0.49 s |
| GraphMAE2 | 1.25 s | 0.45 s | 0.29 s | 31.70 s | 0.62 s | 0.54 s |
| **TopoDIG (Ours)** | 1.04 s | 0.35 s | 0.21 s | 25.30 s | 0.45 s | 0.38 s |

Table 10: Running time analysis of different modules within the TopoDIG on ogbn-arxiv.

| Module | Dowker complex | Persistence image | Training | Inference |
|---|---|---|---|---|
| ogbn-arxiv | 5.0 mins | 1.5 mins | 25.30 seconds | 2.6 seconds |

