# OpenReview forum: "Topology-Guided Graph Pre-training and Prompt Learning on Directed Graphs"
_TMLR — Accepted by TMLR_

### Review · Reviewer_c9rd · 2025-12-03

**Summary Of Contributions:**

This paper studies the graph transfer learning on directed graphs. Specifically, the authors develop a new framework called TopoDIG which consists of several techniques, such as magnetic Laplacian, Dowker complex-based features. Extensive results showcase the effectiveness of TopoDIG for various downstream tasks.
However, the datasets adopted in this paper are not mainstream graph benchmarks, which significantly weakens the technique contributions of this paper.

**Audience:**

Yes

**Audience Explanation:**

Graph Representation learning is an interesting topic in the machine learning community.

**Claims And Evidence:**

No

**Claims Explanation:**

The datasets are unsuitable for evaluating the model performance.

**Requested Changes:**

1.There are some typos, for instance “, e.g., ? lists six different pre-training methods”. Please carefully proofread.

2.More details of the framework should be highlighted in Figure 1.

3.The complexity analysis of the proposed method is required.

4.The real-world datasets are too small. Please select mainstream graph benchmarks, such as Computers, Photo, CS, Physics or datasets in OGB for performance evaluation [1].

[1] DUALFormer: Dual Graph Transformer. ICLR 2025.

---

> ### Author Response · Authors · 2026-01-09
> **Response to Reviewer c9rd**
>
> We thank the reviewer for their thoughtful and constructive feedback. We have revised the manuscript accordingly and address each point in detail below.
>
> **Q1:** The datasets are unsuitable for evaluating the model performance. The real-world datasets are too small. Please select mainstream graph benchmarks, such as Computers, Photo, CS, Physics or datasets in OGB for performance evaluation [1].
>
> **A:**
> As our study primarily focuses on directed graphs, we conducted additional experiments on large-scale directed graph benchmarks under the 3-shot learning setting, namely ogbn-arxiv (169,343 nodes and 1,166,243 edges) and ogbn-Papers100M (111,059,956 nodes and 1,615,685,872 edges).
>
> Based on the results in Tables 1 and 2, our method TopoDIG consistently and substantially outperforms all competing baselines across both few-shot and low-label regimes. On the 3-shot node classification task on ogbn-arxiv (see Table 1), TopoDIG achieves the highest accuracy, surpassing the strongest baseline Gprompt by a clear margin, while dramatically outperforming earlier prompting-based and fine-tuning approaches. This demonstrates the effectiveness of TopoDIG in extremely label-scarce settings on large-scale directed citation graphs. For fine-tuning with limited labeled data (see Table 2), TopoDIG again delivers the best performance under both 1% and 5% label ratios, respectively. Compared to the strongest self-supervised baseline, GraphMAE2, TopoDIG yields consistent gains while maintaining comparable variance, highlighting its superior representation quality and robustness. Overall, these results confirm that incorporating topology-aware representations enables TopoDIG to more effectively leverage structural information, leading to superior performance on large-scale directed graphs.
>
> Table 1. Evaluation results for 3-shot node classification (in \%) on ogbn-arxiv.
> | | ogbn-arxiv |
> |-|-|
> | GPPT | 23.35±0.70 |
> | All-in-one | 16.79±6.81 |
> | Gprompt | 82.07±3.19 |
> | GPF | 77.50±7.23 |
> | TopoDIG (Ours) | **86.78±2.20** |
>
> Table 2. Results (in \%) of fine-tuning the pretrained GNN with 1% and 5% labeled training data on large-scale datasets.
> | | Label ratio 1\% |  Label ratio 5\% |
> |-|-|-|
> | CCA-SSG [1] | 55.68±0.24 | 59.78±0.08 |
> | GRACE [2] | 55.45±0.23 | 59.38±0.15 |
> | BGRL [3] | 55.12±0.23 | 60.40±0.54 |
> | GraphMAE [4] | 58.29±0.15 | 62.00±0.12 |
> | GraphMAE2 [5] | 58.69±0.38 | 62.87±0.64 |
> | TopoDIG (Ours) | **59.42±0.41** | **64.05±0.59** |
>
> [1] Hengrui Zhang, Qitian Wu, Junchi Yan, David Wipf, and Philip S Yu. 2021. From canonical correlation analysis to self-supervised graph neural networks. In NeurIPS.
>
> [2] Yanqiao Zhu, Yichen Xu, Feng Yu, Qiang Liu, Shu Wu, and Liang Wang. 2020. Deep graph contrastive representation learning. arXiv preprint arXiv:2006.04131 (2020).
>
> [3] Shantanu Thakoor, Corentin Tallec, Mohammad Gheshlaghi Azar, Rémi Munos, Petar Veličković, and Michal Valko. 2022. Large-Scale Representation Learning on Graphs via Bootstrapping. In ICLR.
>
> [4] Zhenyu Hou, Xiao Liu, Yukuo Cen, Yuxiao Dong, Hongxia Yang, Chunjie Wang, and Jie Tang. 2022. GraphMAE: Self-Supervised Masked Graph Autoencoders. In KDD.
>
> [5] Hou, Z., He, Y., Cen, Y., Liu, X., Dong, Y., Kharlamov, E. and Tang, J., 2023, April. Graphmae2: A decoding-enhanced masked self-supervised graph learner. In Proceedings of the ACM web conference 2023 (pp. 737-746).
>
>
> **Q2:** There are some typos, for instance “, e.g., ? lists six different pre-training methods”. Please carefully proofread.
>
> **A:**  We have fixed the typos we found during proofreading.
>
> **Q3:** The complexity analysis of the proposed method is required.
>
> **A:**  For the magnetic Laplacian module, the computational complexity is $O(|E| \cdot F_{\ell-1} + |V| \cdot F\_{\ell-1} F_{\ell})$ where $F\_{\ell-1}$ denotes the embedding dimension of each node at the input of layer $\ell$ and $F\_{\ell}$ denotes the embedding dimension of each node at the output of layer $\ell$. The complexity of CNN + pooling topological encoder is $O(\mathcal{K} \cdot P^2 \cdot C_{CNN})$ where $\mathcal{K}$ denotes the number of filtration functions, $P$ denotes the resolution size of the persistence image, and $C_{CNN}$ denotes the interaction between input and output channels in standard convolution operations.

---

> > ### Author Response · Authors · 2026-02-02
> > **Official Comment by Authors for Reviewer c9rd**
> >
> > Dear Reviewer c9rd,
> >
> > We greatly appreciate the time and effort you have dedicated to reviewing our work. We hope that all the points raised in your feedback have been adequately addressed.
> >
> > If there are any remaining concerns or areas requiring further clarification, please do not hesitate to let us know. We would be happy to provide additional details or explanations.
> >
> > Best,
> >
> > Authors

---

### Review · Reviewer_xW9S · 2025-12-16

**Summary Of Contributions:**

This paper aims to fuse topological deep learning with direct graph GNNs + a prompt engineering pre-train and then fine tune approach. Input graph is first subjected to two separate feed-forward graphs which are then concatenated and fed into an MLP. The first track consists of convolutions similar to MagNet (Zhang et al.) The second track relies on constructing a Dowker complex and feeding it through a topological learning representation module. The authors validate their method on Texas/Wisconsins/Cornell as well as on a synthetic Directed SBM model.

While this paper shows promise in utilizing many different types of important information in the input graph, significant weaknesses remain. These include (i) computational complexity is not discussed; (ii) Section 3.4 provides insufficient detail; and it's not clear how the prompt engineering fits into the rest of the architecture. (Should it be in Figure 1 somewhere). This last issue is particularly concerning since that is one of the primary innovations of this work.

**Audience:**

Yes

**Audience Explanation:**

TDL and pretrain/finetune are active and exciting areas of research

**Claims And Evidence:**

Yes

**Claims Explanation:**

Model performance is validated on standard benchmarks. Model components are (mostly) well explained.

**Requested Changes:**

The claim the charge parameter controls the emphases on directionality is not quite true (except asymptotically as q->0). Instead, different q emphasize different digraph motifs. Please rephrase.

The first sentence of 3.1 implies that Zhang et al. introduced the magnetic Laplacian. This is false, they merely integrated it into the GNN framework. Please rephrase.

If definition 1, the delta should be an epsilon.

Something is wrong with the definitions of the Wasserstein distances, there is no minus on the RHS.

More details on prompting for downstream tasks should be given in Section 3.4 in order for the paper to be self contained.

Questions:

Is there a way to get oriented simplices out of the original graph? This might eliminate the need for multiple tracks in the model?

Is there a way to tell if a graph arises from some underlying space with interesting topology e.g., a manifold or degenerate manifold.

In section 3.2, how are the Gaussian kernels defined if the graph doesn't lie in a Euclidean space.

None of the experiments report the running time of the model. This is needed in order to determine whether or not the moderate performance increase is enough to justify increases in computational cost.

Minor:

Abstract: capture -> capturing
Page 1: predicting modeling -> predicting and modeling
Broken reference on page 2 and another one on page 10. Please check throughout.
Page 11 "shwon"

---

> ### Author Response · Authors · 2026-01-09
> **Response to Reviewer xW9S (1/2)**
>
> We thank the reviewer for their thoughtful and constructive feedback. We have revised the manuscript accordingly and address each point in detail below.
>
> **Q1:** The claim the charge parameter controls the emphases on directionality is not quite true (except asymptotically as q->0). Instead, different q emphasize different digraph motifs. Please rephrase.
>
> **A:** Thank you for pointing this out. Indeed, the original phrasing “A tunable "charge" parameter allows control over the emphasis on directionality.” may lead to an impression that larger q values naturally lead to more emphases, which is not quite true. We have rephrased this part to “A tunable "charge" parameter is introduced, which asymptotically allows control over the emphasis on directionality for small parameter values. More generally, different charge parameter values induce sensitivity to different directed subgraph motifs.”
>
> **Q2:** The first sentence of 3.1 implies that Zhang et al. introduced the magnetic Laplacian. This is false, they merely integrated it into the GNN framework. Please rephrase.
> A: Thank you for pointing this out. The original sentence indeed may cause confusion. We have rephrased to say that Zhang et al. “applies the concept of a magnetic Laplacian to directed graph neural networks”.
>
> **Q3:** If definition 1, the delta should be an epsilon.
>
> **A:** We thank the reviewer for the question. We have changed the \delta to \epsilon.
>
> **Q4:** Something is wrong with the definitions of the Wasserstein distances, there is no minus on the RHS.
>
> **A:** We have revised the Wasserstein distance, i.e., $W\_p(Dgm(q), Dgm(s))$ $= (\inf\_{\gamma \in \Gamma}\sum\_{x \in Dgm(q)} ||x - \gamma(x) ||^p)^{\frac{1}{p}}$.
>
>
> **Q5:** More details on prompting for downstream tasks should be given in Section 3.4 in order for the paper to be self contained.
>
> **A:** We thank the reviewer for raising this concern. We will clarify the prompting part more explicitly in the revised version of the paper.
>
> **Q6:** Is there a way to get oriented simplices out of the original graph? This might eliminate the need for multiple tracks in the model?
>
> **A:** We thank the reviewer for this insightful suggestion. Since our setting focuses on directed graphs, an orientation scheme is indeed naturally available at the edge level, and in principle one could attempt to lift the directed graph to oriented simplices (e.g., via directed clique or path complexes) to obtain higher-order oriented structures. However, while edge directions provide local orientation, the induced orientations of higher-dimensional simplices are still not unique and can depend on specific construction rules (e.g., node ordering, transitivity assumptions, or path selection), which may introduce arbitrariness and instability in practice. Moreover, a single oriented simplicial construction tends to entangle orientation-sensitive relational information with orientation-invariant topological properties, potentially limiting modeling flexibility. Our multi-track design is therefore adopted to explicitly disentangle these complementary aspects: the direction-aware track captures orientation-sensitive information inherent to the directed graph, while the topology track focuses on stable higher-order topological signatures that are robust to local orientation choices. This design allows us to fully exploit the available directionality while maintaining robustness and expressiveness, and we will clarify this rationale in the revised manuscript.

---

> > ### Author Response · Authors · 2026-01-09
> > **Response to Reviewer xW9S (2/2)**
> >
> > **Q7:** Is there a way to tell if a graph arises from some underlying space with interesting topology e.g., a manifold or degenerate manifold.
> >
> > **A:** In general, there is no definitive or algorithmic test that can determine whether an arbitrary graph arises from an underlying manifold or a (possibly degenerate) manifold solely based on its graph structure. This is because the mapping from latent spaces to graphs is highly non-injective, and many distinct generative processes including non-geometric ones can lead to identical or indistinguishable graphs. Under additional assumptions (e.g., locality of connections and sufficient sampling density), one can provide evidence that a graph is consistent with an underlying low-dimensional space. Such evidence may include locally low-dimensional geometry, spectral properties of the graph Laplacian consistent with discretized differential operators, or stable topological signatures (e.g., persistent homology) across scales. Note that our work does not rely on explicitly identifying or assuming a smooth manifold structure. Instead, the proposed method is designed to remain applicable even when the underlying space is degenerate, stratified, or only approximately manifold-like. Thus, we view manifold structure as a useful interpretive lens rather than a prerequisite assumption.
> >
> > **Q8:** In section 3.2, how are the Gaussian kernels defined if the graph doesn't lie in a Euclidean space.
> >
> > **A:** We thank the reviewer for the question. The Gaussian kernel is used in the persistence image computation (please see its reference [1]). Specifically, the Gaussian kernel is applied to a persistence diagram to generate a persistence surface.
> >
> > Adams, H., Emerson, T., Kirby, M., Neville, R., Peterson, C., Shipman, P., Chepushtanova, S., Hanson, E., Motta, F. and Ziegelmeier, L., 2017. Persistence images: A stable vector representation of persistent homology. Journal of Machine Learning Research, 18(8), pp.1-35.
> >
> > **Q9:** None of the experiments report the running time of the model. This is needed in order to determine whether or not the moderate performance increase is enough to justify increases in computational cost.
> >
> > **A:** Thank you for raising this issue. To address your concern, we have reported the single training epoch time (in second (s)) of the model on multiple graphs.
> > Table 1.The single training epoch time of TopoDIG on different datasets.
> > | | Texas | Cornell | Wisconsin | ogbn-arxiv
> > |-|-|-|-|-|
> > | Running time | 1.04 s | 0.35 s | 0.21 s | 25.30 s
> >
> > **Q10:** Abstract: capture -> capturing Page 1: predicting modeling -> predicting and modeling Broken reference on page 2 and another one on page 10. Please check throughout. Page 11 "shwon".
> >
> > **A:** We have fixed all typos mentioned here.

---

> > > ### Author Response · Authors · 2026-02-02
> > > **Official Comment by Authors for Reviewer xW9S**
> > >
> > > Dear Reviewer xW9S,
> > >
> > > We greatly appreciate the time and effort you have dedicated to reviewing our work. We hope that all the points raised in your feedback have been adequately addressed.
> > >
> > > If there are any remaining concerns or areas requiring further clarification, please do not hesitate to let us know. We would be happy to provide additional details or explanations.
> > >
> > > Best,
> > >
> > > Authors

---

> > > > ### Comment · Reviewer_xW9S · 2026-02-05
> > > > **Satisfied with revisions**
> > > >
> > > > Thank you for uploading your revision. It has satisfied my concerns.

---

> > > > > ### Author Response · Authors · 2026-02-05
> > > > > **Official Comment by Authors**
> > > > >
> > > > > Dear Reviewer xW9S,
> > > > >
> > > > > Thanks very much for your valuable time and effort in helping us improve our work!
> > > > >
> > > > > Best,
> > > > >
> > > > > Authors

---

### Review · Reviewer_dTdv · 2025-12-24

**Summary Of Contributions:**

The authors propose TopoDIG, a method for pretraining and prompting on directed graphs. The approach combines Magnetic Laplacian convolution (encoding edge directionality via complex phase shift) with topology-aware layers operating on persistence images from Dowker complex filtrations. The motivation—bringing persistent homology-style structure into directed graph representation learning—is interesting and timely. The paper evaluates TopoDIG on 3 WebKB datasets and synthetic Directed Stochastic Block Models (DSBMs).

Key strengths:

- Promising combination of directed GNNs (Magnetic Laplacian) with topological features (Dowker/persistent homology)
- Timely motivation for incorporating topology into directed graph representation learning

Key weaknesses:

- Empirical validation is limited (few real datasets) relative to the breadth of the claims
- Runtime/memory/scaling analysis is missing despite substantial computational components
- Space allocation is imbalanced: the main paper spends too much space on non-novel background relative to the core contributions and experiments

**Additional Comments:**

It is reasonable for this paper to focus primarily on methodology and empirical results rather than new theory. That said, the draft would benefit from a stronger experimental story and clearer positioning. In particular, if you claim to fill a significant gap in the literature, please motivate _why_ jointly modeling directionality and topology during pre-training should improve downstream performance in meaningful settings (and when it should not). The idea is promising, but the current experimental evidence is too limited to support definitive, broad claims.

**Audience:**

Yes

**Audience Explanation:**

The intersection of topological data analysis, directed graphs, and transfer learning is relevant to TMLR's audience. Applying Dowker complexes to directed graph learning is novel and addresses an underexplored area. However, presentation and experimental validation need substantial improvement to realize the potential impact.

**Broader Impact Concerns:**

Broader Impact statement not included. No concerns

**Claims And Evidence:**

No

**Claims Explanation:**

While the paper combines appealing ingredients (Magnetic Laplacian GNNs and Dowker complex-based persistent homology), the current evidence and presentation do not yet support the strongest claims:

**Insufficient Experimental Validation**: Experiments are limited to WebKB (3 small citation networks) and synthetic DSBMs. For a paper positioning itself as a general advance in directed graph learning, broader evaluation on diverse directed graph domains (e.g., molecular, social, biological, traffic) is needed.

**Missing Computational Cost Analysis**: The method includes computationally heavy components (Dowker complexes, persistence images, CNN processing, Magnetic Laplacian operations). Without runtime/memory comparisons and scaling results, it is hard to assess whether the reported accuracy gains justify the added complexity.

**Methodological Clarity Issues**: Several technical details are unclear:

- Integration of node features with the topological CNN is ambiguous (despite $X$ appearing in equation 6)
- Please devote more space to clearly delineating pretraining vs. prompting: tasks/objectives used in each stage, what supervision is used (if any), and which parameters are updated or frozen
- Equation 8 and the surrounding text do not make it clear whether the prompt function is novel or a straightforward application of existing prompting templates to the proposed encoder

**Requested Changes:**

### Critical Changes (Required for Future Acceptance)

1. **Expand Experimental Validation** (Critical): Current experiments are insufficient for claiming state-of-the-art performance:

   - Add new datasets beyond WebKB (molecular, social, biological, transportation networks)

2. **Provide Computational Cost Analysis** (Critical): Include runtime and memory comparisons:

   - Wall-clock time for: (a) Dowker complexes, (b) persistence images, (c) training, (d) inference
   - Memory requirements vs. baselines
   - Scalability: performance degradation with graph size
   - Justify when added complexity is worthwhile

### Additional Changes

3. **Restructure the Methods Section**: The Methods section extensively repeats existing work without highlighting novel contributions:

   - Move detailed Magnetic Laplacian definitions to the appendix (unless diverging from MagNet)
   - Condense the persistent homology stability discussion (well-known theory) or clearly state any paper-specific theoretical contribution; avoid spending substantial space on stability framing without a new result
   - Focus on: (a) combining components, (b) novelty, (c) motivation
   - Use saved space to expand experiments and clarify motivation

4. **Strengthen the Related Work Section**: Several issues need addressing:

   - Quantify DGCN's efficiency concerns and justify choosing MagNet as the directed baseline/starting point
   - Clarify that MoAMa is tailored to molecular graphs, which limits how broadly its conclusions transfer
   - Expand the graph prompting discussion: distinguish paradigms and explain which one you adopt and why
   - Support the claim that prior approaches "fall short" with concrete examples (what directional/topological signals are missing, and why they matter)
   - Add discussion of prior uses of Dowker complexes in machine learning
   - Cover existing TDA work on directed graphs (even if limited)

5. **Provide Use Case Analysis**: Given computational complexity, clarify:

   - Which graph classes benefit most (small molecular vs. large social networks)?
   - When to use TopoDIG versus simpler alternatives
   - Which directed graph properties make topological features valuable

---

> ### Author Response · Authors · 2026-01-09
> **Response to Reviewer dTdv (1/3)**
>
> We thank the reviewer for their thoughtful and constructive feedback. We have revised the manuscript accordingly and address each point in detail below.
>
> **Q1:** Add new datasets beyond WebKB (molecular, social, biological, transportation networks)
>
> **A:** We have conducted additional experiments on large-scale directed graph benchmarks under the 3-shot learning setting, namely ogbn-arxiv (169,343 nodes and 1,166,243 edges) and ogbn-Papers100M (111,059,956 nodes and 1,615,685,872 edges).
>
> Based on the results in Tables 1 and 2, our method TopoDIG consistently and substantially outperforms all competing baselines across both few-shot and low-label regimes. On the 3-shot node classification task on ogbn-arxiv (see Table 1), TopoDIG achieves the highest accuracy, surpassing the strongest baseline Gprompt by a clear margin, while dramatically outperforming earlier prompting-based and fine-tuning approaches. This demonstrates the effectiveness of TopoDIG in extremely label-scarce settings on large-scale directed citation graphs. For fine-tuning with limited labeled data (see Table 2), TopoDIG again delivers the best performance under both 1% and 5% label ratios, respectively. Compared to the strongest self-supervised baseline, GraphMAE2, TopoDIG yields consistent gains while maintaining comparable variance, highlighting its superior representation quality and robustness. Overall, these results confirm that incorporating topology-aware representations enables TopoDIG to more effectively leverage structural information, leading to superior performance on large-scale directed graphs.
>
> Table 1. Evaluation results for 3-shot node classification (in \%) on ogbn-arxiv.
> | | ogbn-arxiv |
> |-|-|
> | GPPT | 23.35±0.70 |
> | All-in-one | 16.79±6.81 |
> | Gprompt | 82.07±3.19 |
> | GPF | 77.50±7.23 |
> | TopoDIG (Ours) | **86.78±2.20** |
>
> Table 2. Results (in \%) of fine-tuning the pretrained GNN with 1% and 5% labeled training data on large-scale datasets.
> | | Label ratio 1\% |  Label ratio 5\% |
> |-|-|-|
> | CCA-SSG [1] | 55.68±0.24 | 59.78±0.08 |
> | GRACE [2] | 55.45±0.23 | 59.38±0.15 |
> | BGRL [3] | 55.12±0.23 | 60.40±0.54 |
> | GraphMAE [4] | 58.29±0.15 | 62.00±0.12 |
> | GraphMAE2 [5] | 58.69±0.38 | 62.87±0.64 |
> | TopoDIG (Ours) | **59.42±0.41** | **64.05±0.59** |
>
> [1] Hengrui Zhang, Qitian Wu, Junchi Yan, David Wipf, and Philip S Yu. 2021. From canonical correlation analysis to self-supervised graph neural networks. In NeurIPS.
>
> [2] Yanqiao Zhu, Yichen Xu, Feng Yu, Qiang Liu, Shu Wu, and Liang Wang. 2020. Deep graph contrastive representation learning. arXiv preprint arXiv:2006.04131 (2020).
>
> [3] Shantanu Thakoor, Corentin Tallec, Mohammad Gheshlaghi Azar, Rémi Munos, Petar Veličković, and Michal Valko. 2022. Large-Scale Representation Learning on Graphs via Bootstrapping. In ICLR.
>
> [4] Zhenyu Hou, Xiao Liu, Yukuo Cen, Yuxiao Dong, Hongxia Yang, Chunjie Wang, and Jie Tang. 2022. GraphMAE: Self-Supervised Masked Graph Autoencoders. In KDD.
>
> [5] Hou, Z., He, Y., Cen, Y., Liu, X., Dong, Y., Kharlamov, E. and Tang, J., 2023, April. Graphmae2: A decoding-enhanced masked self-supervised graph learner. In Proceedings of the ACM web conference 2023 (pp. 737-746).
>
> **Q2:** Provide Computational Cost Analysis: Include runtime and memory comparisons: Wall-clock time for: (a) Dowker complexes, (b) persistence images, (c) training, (d) inference Memory requirements vs. baselines Scalability: performance degradation with graph size Justify when added complexity is worthwhile.
>
> **A:** For the ogbn-arxiv data, the topological preprocessing requires 5.0 minutes for Dowker complex construction and 1.5 minutes for persistence image generation, and is performed once per graph and reused across runs. Training takes 25.30 seconds per epoch and full-graph inference requires 2.6 seconds. During topological preprocessing (Dowker complex construction and persistence image computation), the peak CPU memory usage is around 35 GB. During model training, the GPU memory peak is around 9.0 GB. The resulting topological features are compact and reusable, which requires around 0.5 GB.
>
> | | ogbn-arxiv |
> |-|-|
> | Dowker complex | 5.0 mins
> | Persistence image | 1.5 mins
> | Training | 25.30 seconds
> | Inference | 2.6 seconds

---

> > ### Author Response · Authors · 2026-01-09
> > **Response to Reviewer dTdv (2/3)**
> >
> > **Q3:** Move detailed Magnetic Laplacian definitions to the appendix (unless diverging from MagNet); Condense the persistent homology stability discussion (well-known theory) or clearly state any paper-specific theoretical contribution; avoid spending substantial space on stability framing without a new result; Focus on: (a) combining components, (b) novelty, (c) motivation.
> >
> > **A:** We thank the reviewer for raising this concern. In the revised version, we will restructure the paper to sharpen the focus on our core contributions and improve clarity.
> >
> > **Q4:** Quantify DGCN's efficiency concerns and justify choosing MagNet as the directed baseline/starting point.
> >
> > **A:** Thank you for raising this issue. We have compared the efficiency of DGCN and MagNet by providing the single training epoch time (in second (s)) of the model on multiple graphs. These results quantitatively confirm the efficiency concerns associated with DGCN, particularly for large graphs. MagNet provides a more computationally efficient yet expressive framework for modeling directed graphs, making it a suitable foundation for integrating additional components without incurring prohibitive training costs.
> >
> > Table 3. Efficiency comparison on different datasets.
> > | | Texas | Cornell | Wisconsin | ogbn-arxiv
> > |-|-|-|-|-|
> > | Running time with DGCN | 1.38 s | 0.50 s | 0.32 s | 34.43 s
> > | Running time with MagNet | 1.04 s | 0.35 s | 0.21 s | 25.30 s
> >
> > **Q5:** Clarify that MoAMa is tailored to molecular graphs, which limits how broadly its conclusions transfer.
> >
> > **A:** We thank the reviewer for pointing this out. The MoAMa is specifically designed for molecular graphs, where nodes and edges correspond to atoms and chemical bonds with well-defined physicochemical semantics and constraints. As a result, its modeling assumptions and conclusions are tailored to molecular structure learning and do not directly generalize to arbitrary directed or relational graphs. In contrast, in our work, we target general directed graphs without domain-specific chemical priors, which is applicable to a broader class of problems. We will revise the related work and experimental discussion to clearly delineate this distinction in scope, ensuring that the comparison is interpreted appropriately and that conclusions are not overstated beyond MoAMa’s intended domain.
> >
> > **Q6:** Expand the graph prompting discussion: distinguish paradigms and explain which one you adopt and why.
> >
> > **A:** We thank the reviewer for raising this concern. We will clarify the prompting part more explicitly in the revised version of the paper.
> >
> > **Q7:** Support the claim that prior approaches "fall short" with concrete examples (what directional/topological signals are missing, and why they matter).
> >
> > **A:** We compare our method with those baselines (which fall short in their ability to incorporate directional structure and topological information, the essence of directed graphs) on ogbn-arxiv data.
> > As shown in Table 4, TopoDIG consistently outperforms prior prompt-based baselines on ogbn-arxiv, achieving an absolute improvement of 4.7\% over Gprompt and 9.3\% over GPF in the 3-shot setting. In particular, GPPT and All-in-one treat the graph largely as direction-agnostic or text-dominated, thereby missing edge orientation, i.e., dependent information flow that is critical in citation networks such as ogbn-arxiv. While Gprompt and GPF leverage graph structure, they primarily rely on local neighborhood aggregation, which is insufficient to encode global directional asymmetries (e.g., citation hierarchies) and higher-order topological patterns (e.g., loops). By explicitly modeling directional structure and incorporating persistent topological signals, our TopoDIG captures both how information flows and which multi-scale structures persist which leads to improved robustness in the low-shot regime.
> >
> > Table 4. Evaluation results for 3-shot node classification (in \%) on ogbn-arxiv.
> > | | ogbn-arxiv |
> > |-|-|
> > | GPPT | 23.35±0.70 |
> > | All-in-one | 16.79±6.81 |
> > | Gprompt | 82.07±3.19 |
> > | GPF | 77.50±7.23 |
> > | TopoDIG (Ours) | **86.78±2.20** |

---

> > > ### Author Response · Authors · 2026-01-09
> > > **Response to Reviewer dTdv (3/3)**
> > >
> > > **Q8:** Add discussion of prior uses of Dowker complexes in machine learning; Cover existing TDA work on directed graphs (even if limited).
> > >
> > > **A:** We thank the reviewer for raising this concern. We will add this discussion in the revised version of the paper.
> > >
> > > **Q9:** Given computational complexity, clarify: Which graph classes benefit most (small molecular vs. large social networks)? When to use TopoDIG versus simpler alternatives;
> > >
> > > **A:** We thank the reviewer for raising this important practical question. TopoDIG is most advantageous for medium-to-large directed graphs where both edge directionality and higher-order topological structure play a meaningful role, such as citation networks (e.g., ogbn-arxiv). In these settings, global directional asymmetries and persistent multi-scale structures (e.g., long-range flows, cycles, hierarchical patterns) cannot be adequately captured by purely local or direction-agnostic models.

---

> > > > ### Author Response · Authors · 2026-02-02
> > > > **Official Comment by Authors for Reviewer dTdv**
> > > >
> > > > Dear Reviewer dTdv,
> > > >
> > > > We greatly appreciate the time and effort you have dedicated to reviewing our work. We hope that all the points raised in your feedback have been adequately addressed.
> > > >
> > > > If there are any remaining concerns or areas requiring further clarification, please do not hesitate to let us know. We would be happy to provide additional details or explanations.
> > > >
> > > > Best,
> > > >
> > > > Authors

---

> ### Author Response · Authors · 2026-02-05
> **Official Comment by Authors**
>
> Dear Reviewer dTdv,
>
> We have revised the paper in response to all reviewers' comments, and the updated changes are highlighted in blue in the revised manuscript. We welcome any further questions or discussions and appreciate the opportunity to enhance our work.
>
> Thank you once again for your valuable time and effort!
>
> Best,
>
> Authors

---

### Decision · Action_Editor_cR1M · 2026-02-23

**Recommendation:** Accept as is

**Audience:**

Yes

**Audience Explanation:**

The authors propose a graph-learning method for directed graphs that combines magnetic Laplacian convolution with topology-aware layers operating on persistence images. This is of interest for the graph ML community.

**Claims And Evidence:**

Yes

**Claims Explanation:**

The authors perform experiments on multiple graph datasets (real and simulated) and perform comparisons to multiple existing graph-learning methods. Several ablation experiments help clarify the effect of individual components.